# Hypergraph-based connectivity measures for signaling pathway topologies

**Nicholas Franzese**[1,2,3¤], **Adam Groce**[2], **T. M. Murali**[3,4], **Anna Ritz**[1]*

**1** Biology Department, Reed College, Portland, Oregon, United States of America, **2** Computer Science Department, Reed College, Portland, Oregon, United States of America, **3** Department of Computer Science, Virginia Tech, Blacksburg, Virginia, United States of America, **4** ICTAS Center for Systems Biology of Engineered Tissues, Virginia Tech, Blacksburg, Virginia, United States of America

¤ Current address: Department of Computer Science, University of Maryland College Park, College Park, Maryland, United States of America

* aritz@reed.edu

**Data Availability Statement:** All data was gathered from PathwayCommons (http://www.pathwaycommons.org/) and the STRING database (http://version10.string-db.org/). Code and details are available on GitHub

## Abstract

Characterizing cellular responses to different extrinsic signals is an active area of research, and curated pathway databases describe these complex signaling reactions. Here, we revisit a fundamental question in signaling pathway analysis: are two molecules "connected" in a network? This question is the first step towards understanding the potential influence of molecules in a pathway, and the answer depends on the choice of modeling framework. We examined the connectivity of Reactome signaling pathways using four different pathway representations. We find that Reactome is very well connected as a graph, moderately well connected as a compound graph or bipartite graph, and poorly connected as a hypergraph (which captures many-to-many relationships in reaction networks). We present a novel relaxation of hypergraph connectivity that iteratively increases connectivity from a node while preserving the hypergraph topology. This measure, *B*-relaxation distance, provides a parameterized transition between hypergraph connectivity and graph connectivity. *B*-relaxation distance is sensitive to the presence of small molecules that participate in many functionally unrelated reactions in the network. We also define a score that quantifies one pathway's downstream influence on another, which can be calculated as *B*-relaxation distance gradually relaxes the connectivity constraint in hypergraphs. Computing this score across all pairs of 34 Reactome pathways reveals pairs of pathways with statistically significant influence. We present two such case studies, and we describe the specific reactions that contribute to the large influence score. Finally, we investigate the ability for connectivity measures to capture functional relationships among proteins, and use the evidence channels in the STRING database as a benchmark dataset. STRING interactions whose proteins are *B*-connected in Reactome have statistically significantly higher scores than interactions connected in the bipartite graph representation. Our method lays the groundwork for other generalizations of graph-theoretic concepts to hypergraphs in order to facilitate signaling pathway analysis.

(https://github.com/annaritz/pathway-connectivity). This repo contains instructions for parsing PathwayCommons v10 (https://github.com/annaritz/pathway-connectivity/tree/master/data) and STRING v11.0 (https://github.com/annaritz/pathway-connectivity/tree/master/data/STRING).

**Funding:** This work is supported by the National Science Foundation under grants DBI-1750981 (to PI Anna Ritz) and CCF-1617678 (to PI T.M. Murali). The funders had no role in study design, data collection and analysis, decision to publish, or preparation of the manuscript.

**Competing interests:** The authors have declared that no competing interests exist.

## Author summary

Signaling pathways describe how cells respond to external signals through molecular interactions. As we gain a deeper understanding of these signaling reactions, it is important to understand how molecules may influence downstream responses and how pathways may affect each other. As the amount of information in signaling pathway databases continues to grow, we have the opportunity to analyze properties about pathway structure. We pose an intuitive question about signaling pathways: when are two molecules "connected" in a pathway? This answer varies dramatically based on the assumptions we make about how reactions link molecules. Here, examine four approaches for modeling the structural topology of signaling pathways, and present methods to quantify whether two molecules are "connected" in a pathway database. We find that existing approaches are either too permissive (molecules are connected to many others) or restrictive (molecules are connected to a handful of others), and we present a new measure that offers a continuum between these two extremes. We then expand our question to ask when an entire signaling pathway is "downstream" of another pathway, and show two case studies from the Reactome pathway database that uncovers pathway influence. Finally, we show that the strict notion of connectivity can capture functional relationships among proteins using an independent benchmark dataset. Our approach to quantify connectivity in pathways considers a biologically-motivated definition of connectivity, laying the foundation for more sophisticated analyses that leverage the detailed information in pathway databases.

## Introduction

A major effort in molecular systems biology is to identify signaling pathways, the networks of reactions that link extracellular signals to downstream cellular responses. Computational representations of signaling pathways have increased in complexity, moving from gene sets to pairwise interactions in the past two decades [1]. Graphs are common representations of protein networks, where nodes are proteins and edges represent pairwise interactions between two proteins. While graph representations have been useful for pathway analysis [2–5] and disease-related applications [5–7], the limitations of graphs for representing biochemical reactions are well recognized [8–12].

Many pathway databases [13–20] have adopted reaction-centric signaling pathway formats such as the Biological Pathway Exchange (BioPAX) [21], which provides more mechanistic information about the interactions. As reaction-centric information has become available, many modeling frameworks have been proposed to overcome the limitations of graphs for analyzing signaling pathway structure [8, 9, 22, 23]. Compound graphs [24, 25] and metagraphs [8] aim to represent protein complexes and hierarchical relationships among molecular entities in the cell. Factor graphs [26] have been used to infer pathway activity from heterogeneous data types. Hypergraphs [27, 28] are generalizations of directed graphs that allow multiple inputs and outputs, and their realization as a model for signaling pathways is emerging [9, 11, 29]. Other models such as Petri nets [30] and logic networks [31, 32] move away from structural network analysis and towards discrete dynamic modeling. Many of these modeling frameworks have an underlying bipartite graph structure.

These new representations have improved fidelity to the underlying biology of signaling reactions but also exhibit increased mathematical and algorithmic complexity. In this light, we examine a fundamental topological concept: when are two molecules "connected" in a signaling pathway? Defining and establishing connectivity is the first step to determining

downstream or upstream elements of a molecule, which may indicate the influence of its activity or the effect of its perturbation. Connectivity is also central to computational methods for identifying potential off-target effects, determining pathway crosstalk, and computing portions of pathways that may be altered in disease.

We first begin by considering existing connectivity measures on four distinct representations of the Reactome pathway database [13, 14]. We demonstrate that these measures range from highly permissive (e.g., path-based connectivity in graphs) to very restrictive (e.g., connectivity in directed hypergraphs), depending on the representation. Thus, two molecules may be "connected" in one representation of a pathway and "disconnected" in another representation. We then introduce *B*-relaxation distance, a parameterized relaxation of connectivity that offers a tradeoff between the permissive and restrictive representations. We show that this new version of connectivity uncovers more subtle structures within the pathway topologies than previous measures, and is sensitive to the presence of small molecules that participate in many reactions. We then consider 34 Reactome signaling pathways and use *B*-relaxation distance to capture the downstream influence of one pathway on another. *B*-relaxation distance allows us to gradually relax the connectivity constraints in hypergraphs, calculating pathway influence at each step. The directed graph representation of Reactome is too highly connected to enable the discovery of such relationships, and while these relationships appear in the bipartite graph representation, they only emerge further away from the upstream "source" pathway. We describe two case studies of pathway influence that we recovered, and describe the specific reactions that contribute to the large influence score. Finally, we use the STRING database to benchmark proteins that are connected in Reactome as a signal of functional relationships, and show that stricter measures of connectivity are enriched for higher-scoring functional interactions across multiple STRING evidence channels.

## Results

### Connectivity analysis using established traversal algorithms

We considered four established directed representations of signaling pathway topology and their associated measures of connectivity (Fig 1). Directed graphs describe relationships among molecules (proteins, and small molecules), while the other models describe relationships among entities that include proteins and small molecules, their modified forms, protein complexes, and protein families. Please refer to the Methods for full details about these representations, including how they are built.

1. **Directed graphs** represent molecules as nodes and interactions as pairwise edges. Interactions may be directed (such as regulation) or bidirected (such as physical binding). We use a Breadth First Search (BFS) traversal to calculate the distance of a source node to all other nodes in the graph.

2. **Compound graphs** represent interactions between pairs of nodes, which may be molecules or groups of molecules (e.g., protein complexes or protein families). We use a previously-established algorithm that traverses the BioPAX structure as a compound graph according to biologically meaningful rules [25]. These rules are encoded within the BioPAX file format, and are described in more detail in the Methods under "Compound graph connectivity." The method, CommonStream, takes as input a limit on the distance from the source node which defines a search boundary. We run CommonStream and iteratively increase the search limit until no new entities are returned.

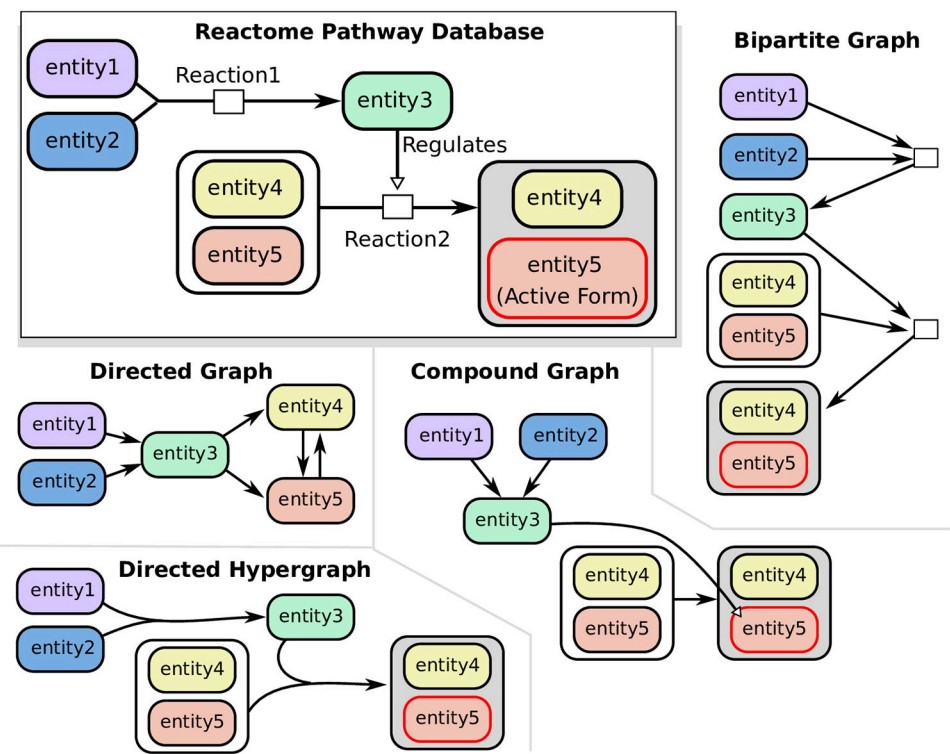

**Fig 1. Representations of two toy reactions as directed graphs, compound graphs, directed hypergraphs, and bipartite graphs.** In this work, we use "directed hypergraphs" and "hypergraphs" interchangeably.

3. **Bipartite graphs** contain two types of nodes: entity nodes and reaction nodes. Each biochemical reaction has an associated reaction node, whose incoming edges are connected to reactants and whose outgoing edges are connected to products. We use BFS to calculate the distance of a source node to all other nodes in the bipartite graph. Like the directed graph, there are no biologically-inspired rules that govern which edges can be traversed.

4. **Directed hypergraphs** represent reactions with many-to-many relationships, where each hyperedge $e = (T_e, H_e)$ has a set of entities in the tail $T_e$ and a set of entities in the head $H_e$ (here, the tail denotes the hyperedge inputs and the head denotes the hyperedge outputs). We adopt a definition of connectivity called *B-connectivity* that requires all the nodes in the tail of a hyperedge to be visited before it can be traversed [28]. This definition has a natural biological meaning in reaction networks: *B-connectivity* requires that all reactants of a reaction must be present in order for any product of that reaction is reachable [11, 28]. Unlike the compound graph rules, *B-connectivity* describes the strictest version of connectivity, and it is the only rule used to traverse hyperedges. We use an integer linear program to compute the number of hyperedges in the shortest hyperpath between all pairs of nodes [29, 33].

We converted the Reactome pathway database to each of the four representations in an effort to determine if they agreed on connectivity (Table 1). The directed graph has fewer nodes than other representations because it includes only proteins, but contains far more edges due to large number of "in-complex-with", "catalysis-precedes", and "controls-state-change-of" binary relations in the SIF file (S2 Table). The hypergraph, compound graph, and

**Table 1. Reactome database representations.**

|  | Directed Graph | Compound Graph | Bipartite Graph | Hypergraph |
|---|---|---|---|---|
| # Nodes | 12,086 | 19,650 | 30,775 | 19,650 |
| # Edges/Hyperedges | 444,204 | 38,218 | 45,155 | 11,125 |

bipartite graph, which are built from the BioPAX files, have more nodes than the graph since they include protein complexes, families and modified forms as distinct entities. However, since each hyperedge is a multi-way relationship, the number of hyperedges is smaller than the number of edges in directed graphs. The compound graph and bipartite graph contain more edges than the hypergraph since they describe relationships among entities using pairwise edges.

The directed graph representation of Reactome is relatively well-connected, with about 80% of the node pairs in the graph connected by a path of length 5 or fewer (Fig 2A). We then considered reachability from each node separately, and found that, nearly 90% of the nodes reached over 80% of the network due to the large number of edges (Fig 2B). For the other representations, we surveyed the same 19,650 entities representing proteins, small molecules, complexes, and families. These representations are much sparser, where only 30–40% of the node pairs in the compound graph and bipartite graph are reachable (Fig 2A). Two-thirds of

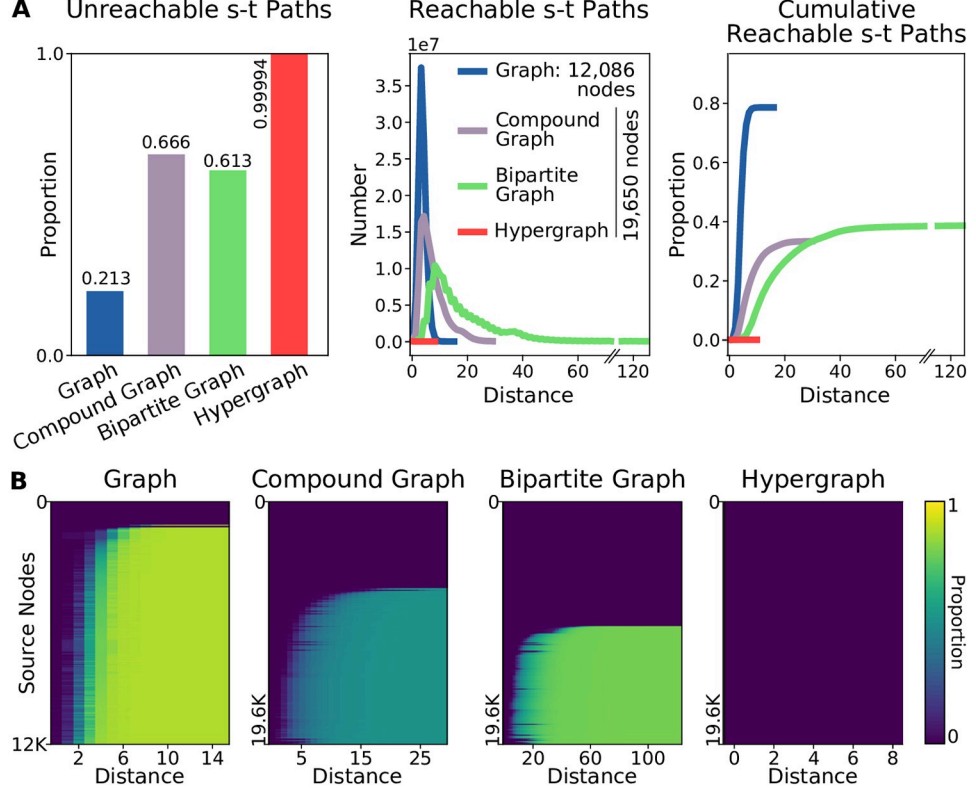

**Fig 2. Reactome connectivity across pathway representations (directed graph BFS, compound graph traversal [25], bipartite graph BFS, and hypergraph *B*-connectivity [28]).** (A) The number of node pairs in each representation that (left) are not connected by a path, (middle) are connected by a path of distance *k*, and (right) are connected by a path with at most distance *k*. (B) Heatmaps showing the proportion of nodes that are reached with distance *k* for every node surveyed.

the nodes in the compound graph representation reach 50% of the network while half the nodes in the bipartite graph representation reach about 40% of the network (Fig 2B). In the hypergraph representation, only five of the nodes are connected to more than 20 others in terms of $B$-connectivity, and most of the nodes cannot reach any others (Fig 2A–2B). In hypergraphs, the $B$-connectivity requirement of visiting all nodes in the tail of a hyperedge before traversal is overly strict for Reactome's topology.

## $B$-relaxation distance on hypergraphs

Connectivity in four different representations of Reactome largely exhibits an all-or-nothing behavior: nodes are either connected to very few or a large fraction of all other nodes. Further, while $B$-connectivity is a powerful definition of connectivity, it is too strict to be useful for Reactome. We introduce $B$-relaxation distance, a parameterized relaxation of hypergraph $B$-connectivity that naturally bridges the gap between $B$-connectivity in directed hypergraphs and connectivity in bipartite graphs. When we consider the connectivity from a node $v$ in the hypergraph, nodes with a $B$-relaxation distance of 0 from $v$, denoted $B_0$, are exactly the nodes that are $B$-connected to $v$. Nodes with a $B$-relaxation distance of 1 ($B_1$) allows one hyperedge to be freely traversed, lifting the restriction that all nodes in the tail must be visited in order to traverse the hyperedge. In general, nodes with a $B$-relaxation distance of $k$ ($B_k$) require $k$ hyperedges to be freely traversed. For shorthand, we will denote $B_{\leq k}$ to be the set of nodes with a $B$-relaxation distance from a source node of at most $k$. A formal definition and efficient algorithms for computing $B$-relaxation distance appear in the Methods).

We computed the $B$-relaxation distance from every node in the hypergraph to every other node and plotted $|B_{\leq k}|$ for different values of $k$ (Fig 3A). The first column ($k = 0$) is the number of $B$-connected nodes for each source, a histogram of which is shown in Fig 2B. The last column ($k = 49$) corresponds to the other extreme: for each source node, we display the number of nodes that are $B$-connected to the source while requiring that only one node in the tail of a hyperedge needs to be connected to the source for us to determine that every node in the head of the hyperedge is reachable from the source. The nodes reached for such a large value of $k$ for each source are exactly the nodes that are connected to the source in the bipartite graph

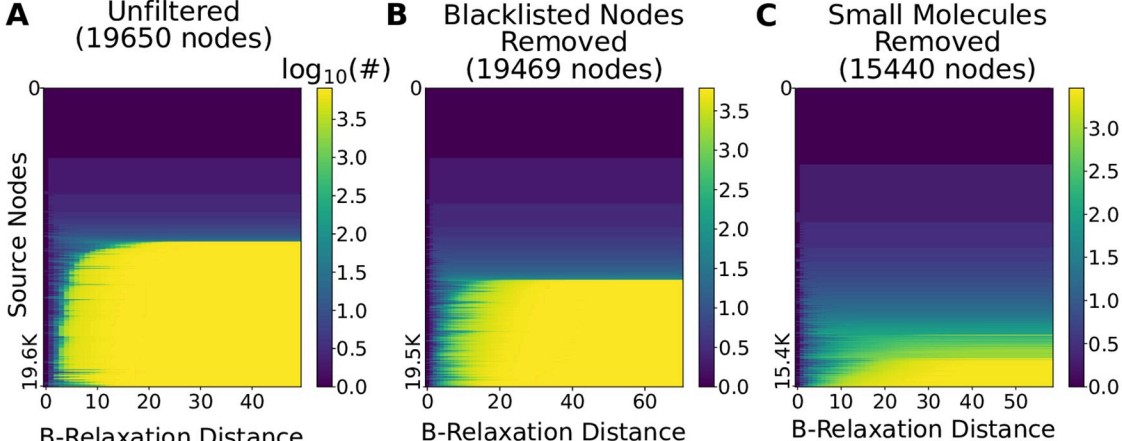

**Fig 3. $B$-relaxation distance survey from each node in the hypergraph.** Heatmaps show the number of nodes $|B_{\leq k}|$ in the $B_k$-connected set from each source node (rows) for values of $k$ (columns) in the hypergraph. (A) $B$-relaxation distance for full hypergraph, (B) $B$-relaxation distance for hypergraph with blacklisted nodes removed [19], and (C) $B$-relaxation distance for hypergraph with small molecules and three highly-connected entities (cytosolic Ubiquitin, nuclear Ubiquitin, and the Nuclear Pore Complex) removed.

representation (Fig 2B). Again, we observe the nodes are divided into two sets: the top blue half are nodes that are connected to very few others and the bottom yellow half are the nodes that are connected to about 40% of the bipartite graph (Fig 3A and S1A Fig).

The nodes in the bottom half of Fig 3A exhibited a transition from reaching very few nodes (blue) to reaching many nodes (yellow). The rapidity of this transition suggested that a small number of nodes may be responsible for it. We hypothesized that these nodes may be small molecules, e.g., ATP, water, sodium and potassium ions, that participate in a vast number of reactions that are functionally unrelated. PathwayCommons reports ubiquitous molecules in their database ("blacklisted nodes" [19]), and 155 of these appear in the hypergraph. Removing these molecules from the hypergraph reduces the number nodes that are connected to many others (Fig 3B). However, we found that small molecules remained even after removing these "blacklisted" nodes. Instead of using the PathwayCommons list, we pruned the hypergraph by removing the 2,778 nodes labeled as small molecules by Reactome, as well as three other highly-connected entities (cytosolic Ubiquitin, nuclear Ubiquitin, and the Nuclear Pore Complex). In total, we altered 5,180 hyperedges by removing these 2,781 entities, resulting in a filtered hypergraph with 15,440 nodes and 8,773 hyperedges. In this hypergraph, even fewer nodes are connected to many others, and the transition from low-to-high connectivity is more gradual across different source nodes (Fig 3C). In contrast, removing PathwayCommons "blacklisted" nodes and small molecules from the directed graph representation changed the distribution very little, suggesting that small molecules played only a minor role in the the high level of connectivity in directed graphs (S1 Fig).

From these results, we concluded that we had a promising definition of parameterized distance that allowed us to relax the strict assumptions posed by *B*-connectivity, and a hypergraph where reachability was not affected by ubiquitous molecules that participate in many reactions. For the remainder of this study, we use the hypergraph with all small molecules removed (Fig 3C).

## Pathway influence across Reactome

While the entire Reactome pathway database appears to be poorly connected in the hypergraph representation, this determination comes from treating individual nodes as sources. We wished to leverage Reactome's pathway annotations to understand how *pathways* are connected in the hypergraph according to *B*-relaxation distance. We identified 34 signaling pathways in Reactome (see "Data formats and representations" in the Methods section and S1 Table) and considered the relationship between pairs of pathways within the hypergraph. When we computed the overlap of the members within each pair of pathways, we found that some pathway pairs already shared nearly all their members (Fig 4A). For example, the normalized overlap between DAG/IP3 signaling and GPCR signaling is 0.9; DAG and IP3 are second messengers in the phosphoinositol pathway, which is activated by GPCRs. The next largest scores are 0.62 and 0.73 between Insulin Receptor signaling and Insulin-like Growth Factor 1 Receptor (IGF1R) signaling. Other growth factor pathways have moderate overlap (e.g., the overlaps among EGFR, ERBB2, and ERBB range from 0.24 to 0.32).

Our aim is to quantify how well a source pathway *S* can reach a target pathway *T* by finding pathway pairs where *T* is "downstream" of *S*. Since we wish to find a directed relationship between pathways, we should ignore the initial overlap between their member sets $P_S$ and $P_T$. Thus, we developed a score that measures how many additional members of *T* may be reached when computing the *B*-relaxation distance from *S*, after accounting for the initial overlap and the total of number of elements that are reached from *S*.

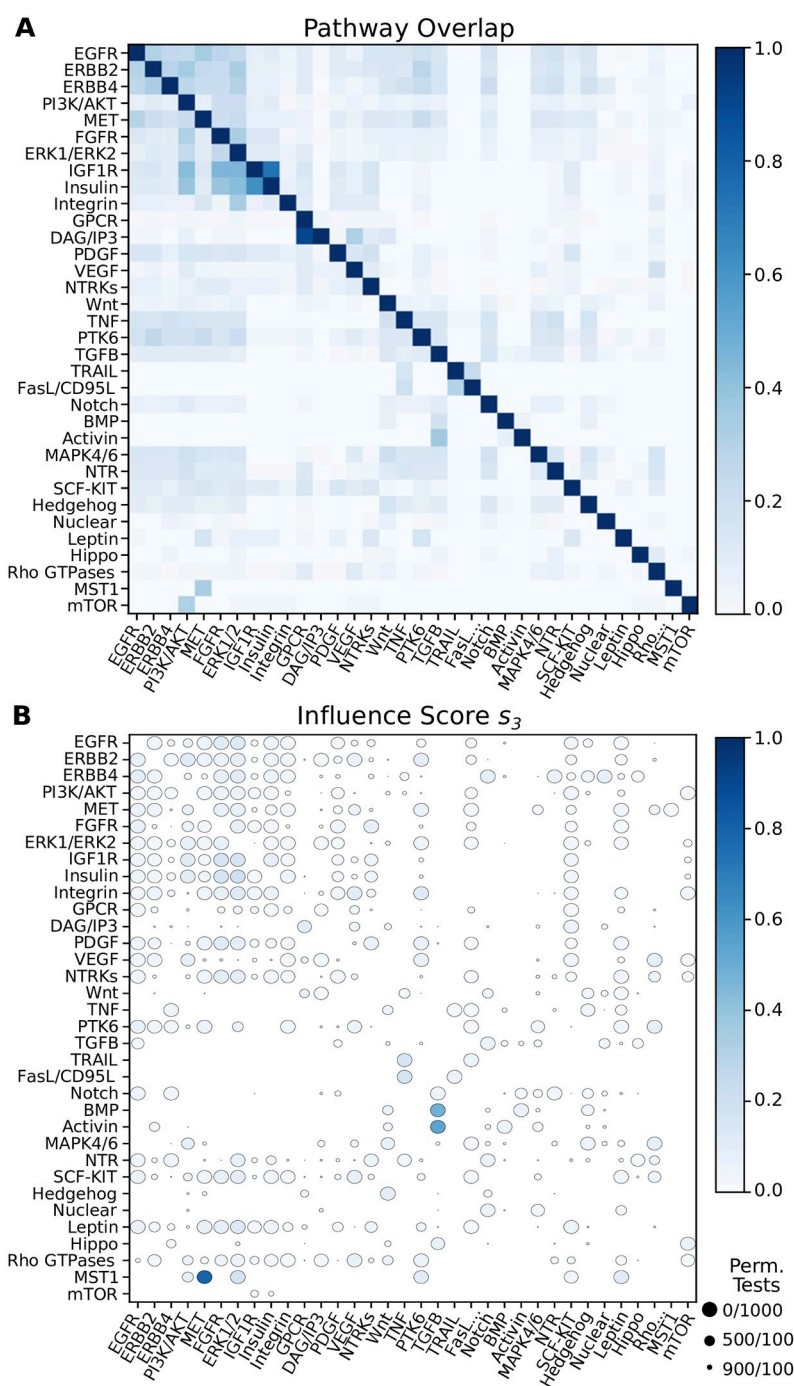

**Fig 4. Overlap and influence of 34 Reactome signaling pathways.** Rows indicate the source pathway $P_S$ and columns indicate the target pathway $P_T$. **(A)** Node overlap of pathway members (normalized by the size of $P_S$). **(B)** Influence scores for $s_3$. Circle size denotes the number of permutations that have scores equal to or greater than the observed influence score.

We define the *influence score* $s_k(S, T)$ of the source pathway $S$ with members $P_S$ on target pathway $T$ with members $P_T$ for $B$-relaxation distance up to $k$ (denoted $B_{\leq k}$) as follows:

$$s_k(S, T) = \frac{|(B_{\leq k}(P_S) \cap P_T) \setminus (P_S \cap P_T)|}{|B_{\leq k}(P_S) \setminus (P_S \cap P_T)|}. \tag{1}$$

This score makes use of the *pathway overlap* between $S$ and $T$ ($P_S \cap P_T$). The numerator counts the number of nodes in $T$ that are reached in the set $B_{\leq k}(P_S)$ that are not already in $P_S$. The denominator counts the total number of nodes that are reached in $B_{\leq k}(P_S)$ that are not in the pathway overlap. Pathway pairs with a large initial overlap are penalized in this score, allowing more subtle patterns to emerge. Moreover, this score penalizes a pathway $P_S$ that reaches many nodes indiscriminately.

We assessed the significance of the influence score for each pathway pair by conducting a permutation test. The permutation test shuffles the node memberships of the pathways while fixing the initial pathway overlap shown in Fig 4A. This is accomplished by using a degree-preserving edge swap on a bipartite graph that represents all possible overlapping sets among the 34 pathway pairs; see S2 Fig for more details. We computed $s_k$ for every pair of Reactome signaling pathways for every value of $k$ (S3 Fig and S1 File). We say a pathway pair's influence score is significant if there are no permutation tests (out of 1,000) with a score greater than or equal to the observed score. There are three significant pairs that exhibit a large influence score for $k = 3$ (large, dark circles in Fig 4B): (a) the Mst1 pathway's influence on MET signaling ($s_3 = 0.79$), (b) the Activin pathway's influence on TGFβ signaling ($s_3 = 0.54$), and (c) the BMP pathway's influence on TGFβ signaling ($s_3 = 0.48$). These pathway pairs have significant influence scores (large circles) for $k = 3$ in the graph and bipartite graph representations (S4 and S5 Figs); however the score is not as large in the graph representation (Table 2). While the influence scores in the bipartite graph mirror those in the hypergraph, they are achieved at a much larger $k$. This is because the the bipartite graph distance (via BFS) conveys a different notion of distance than $B$-relaxation distance, which counts the number of relaxations required to reach entities. We discuss these pathway pairs in two case studies: Mst1 and MET signaling followed by Activin/BMP, and TGFβ signaling.

**Mst1 pathway influence on MET signaling.** Using Macrophage-stimulating Protein 1 (Mst1) as the source pathway $S$, we computed the overlap of the other 33 pathways with $B_{\leq k}$ as $k$ increases (Fig 5A). The largest influence score that we observed across all pathway pairs was 0.79 at $k = 3$ for Mst1 to MET signaling, which indicates that almost all the nodes downstream of Mst1 for $k = 3$ are MET pathway members. For $k = 10$, the set $B_{\leq k}$ contains many ERK1/ERK2 or PI3K/AKT pathway members; however, they comprise a relatively small portion of the total number of nodes in $B_{\leq k}$. The same figure for the bipartite graph representation shows that about the same nodes are recovered at the largest influence score, but this comes at $k = 12$ (S6 Fig).

Fig 5A suggested that the Mst1 pathway may influence the MET pathway. An inspection of the literature and the topology of the nodes in $B_{\leq k}$ from the Mst1 pathway as the source lent

**Table 2. Largest influence scores across all $B$-relaxation distance values for three pathway pairs in Reactome.** The $s_k$ values in bold are the largest influence scores for all pathway pairs for all values of $k$.

| Source & Target Pathways | Directed Graph | Bipartite Graph | Hypergraph |
|---|---|---|---|
| Signaling by Mst1 → Signaling by MET | $s_2 = 0.33$ | $\mathbf{s_{12} = 0.78}$ | $\mathbf{s_3 = 0.79}$ |
| Signaling by Activin → Signaling by TGFβ | $\mathbf{s_2 = 0.36}$ | $s_4 = 0.60$ | $s_2 = 0.59$ |
| Signaling by BMP → Signaling by TGFβ | $s_2 = 0.28$ | $s_{10} = 0.47$ | $s_3 = 0.48$ |

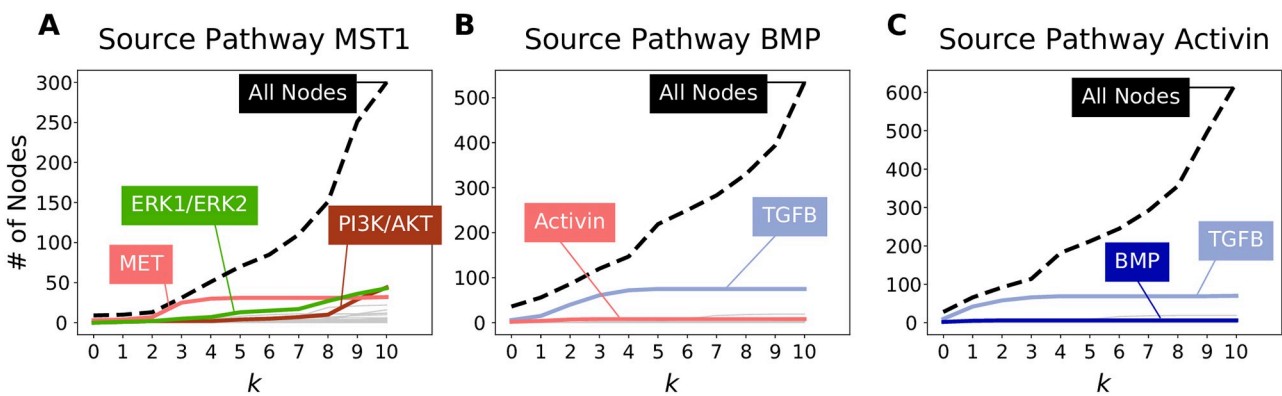

**Fig 5. A single pathway's influence on downstream pathways using B-connectivity.** Shown are the influence of (A) signaling by Mst1, (B) signaling by BMP, and (C) signaling by Activin. The dashed black line indicates the number of nodes in the source pathway's $B_{\leq k}$ for different values of $k$. There is one line for each of the 33 other target pathways denoting the number of members that appear in $B_{\leq k}$, with selected pathways highlighted in bold.

support to this hypothesis. Mst1 is produced in the liver and is involved in organ size regulation [34, 35]. Mst1 acts like a hepatocyte growth factor and has been established as a tumor suppressor gene for heptacellular carcinoma [35]. MET, also known as hepatocyte growth factor (HGF) receptor, is a receptor tyrosine kinase that promotes tissue growth in developmental, wound-healing, and cancer metastasis [36]. Mst1, on the other hand, binds to Mst1R (also known as RON), which is a member of the MET family. Both MET and Mst1R have been shown to have similar downstream effects and can trans-phosphorylate when active [37]. Upon inspection of the reactions that involved the nodes $B_{\leq 3}$, we found that Hepsin (HPN) was involved in forming both the Mst1 dimer and HGF dimer (Fig 6). This protease is known to cleave both pro-Mst1 and pro-HGF into active Mst1 and HGF [38]. The hypergraph also emphasizes the fact that the nodes that are in $B_{\leq k}$ but are not in the MET pathway involve STAT regulation in different cellular compartments. The computed pathway influence (observed as an enrichment of stars in Fig 6 in the regions named $B_0$, $B_1$, $B_2$, and $B_3$) is due to HPN's role in activating the ligands responsible for both Mst1 signaling and MET signaling. Fig 6 also displays the nodes in $B_4$. The high prevalence of nodes that are not in the Met pathway (circles) in this region reinforces the fact that the influence of the Mst1 pathway on the Met pathway is the largest for $k = 3$.

## Activin and BMP influence on TGF$\beta$ signaling

Following the influence score for Mst1 and MET pathways, the next three largest scores across all pathway pairs and all values of $k$ were for the Activin pathway on TGF$\beta$ signaling and the Bone Morphogenic Protein (BMP) pathway on TGF$\beta$ signaling (Table 2). The pattern of $s_k$ values for Activin and TGF$\beta$ were strikingly similar to the trends for BMP and TGF$\beta$ pathways; for both Activin and BMP, TGF$\beta$ was the only target pathway that received a large influence (Fig 5B and 5C). Even though Activin, BMP, and TGF$\beta$ are all known ligands of the TGF$\beta$ superfamily, our analysis demonstrates that the Activin and BMP pathways are upstream of the TGF$\beta$ pathway. The TGF$\beta$ superfamily regulates processes involved in proliferation, growth, and differentiation through both SMAD-dependent and SMAD-independent signaling [39]. TGF$\beta$, Activin, and BMP phosphorylate different SMAD proteins by forming dimers and binding to receptor serine/threonine kinases. TGF$\beta$ binds to TGF$\beta$ Receptor II (TGFBR2), which forms a homodimer with TGFBR1 and activates SMAD2 and SMAD3. Activin also phosphorylates SMAD2 and SMAD2 through binding and activation of the Activin A receptor

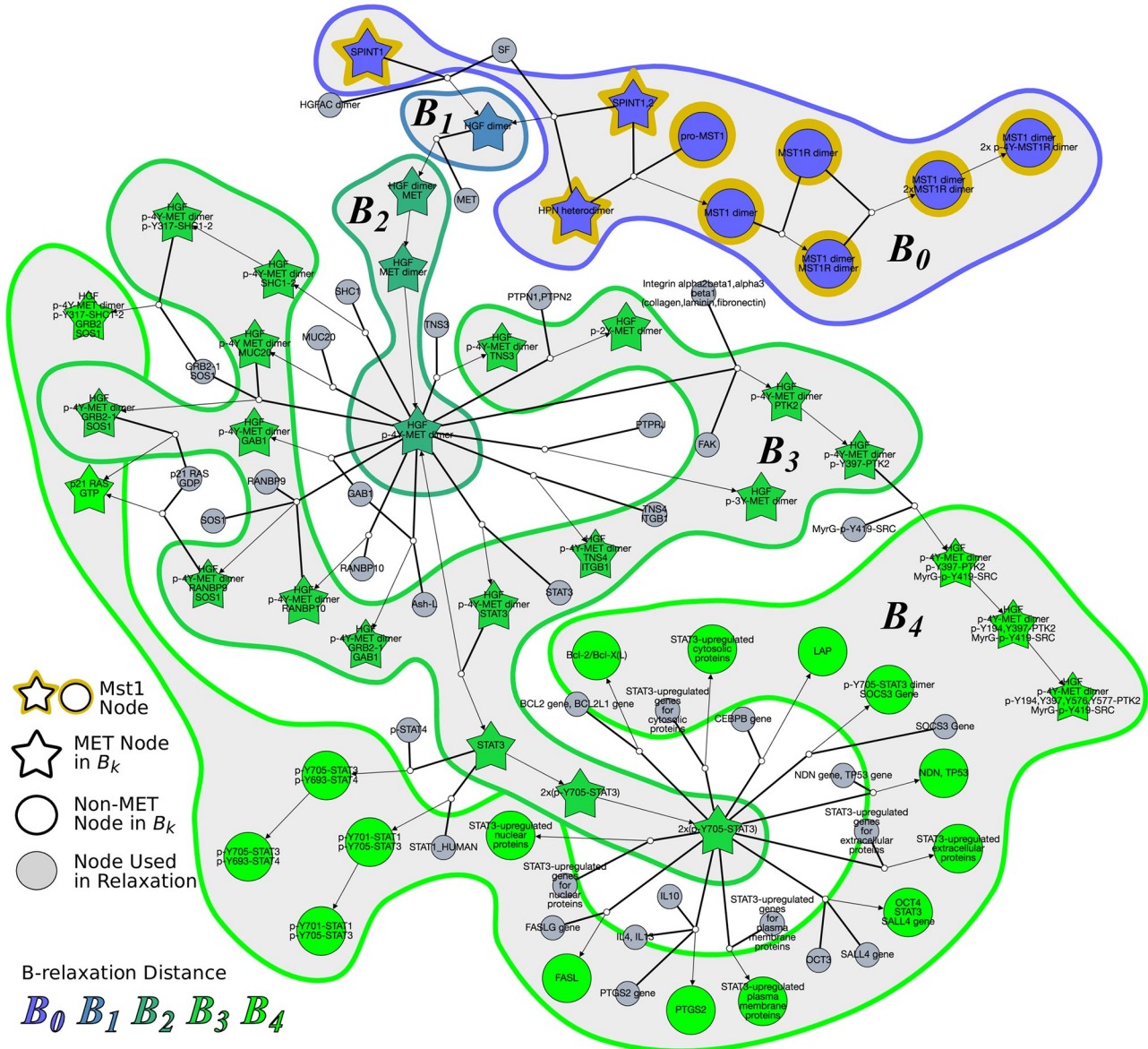

**Fig 6. Hyperedges traversed to compute $B_0$, $B_1$, ..., $B_4$ from source pathway Mst1.** Node colors represent $B$-relaxation distance from $k = 0$ ($B_0$, blue) to $B_4$ (bright green). Gray nodes are entities that are not in $B_k$ but are involved in traversed hyperedges. Star-shaped nodes are members of the MET pathway. This network is available on GraphSpace at http://graphspace.org/graphs/26755?user_layout=6707.

(ACVR). BMP, on the other hand, phosphorylates SMAD1, SMAD5, and SMAD8 through BMP receptor activation. The hypergraph that shows the nodes in $B_{\leq 3}$ from Activin consists of different components and many cycles that denote reuse of SMADs (S6 Fig). The hypergraph suggests that the influence of Activin on TGF$\beta$ does not begin at the ligand, but rather at the activation of SMAD proteins.

## Benchmarking functional relationships using STRING evidence channels

Under any definition of connectivity, two connected proteins may have a functional relationship rather than a physical one. Thus, we sought to benchmark pairs of proteins deemed to be

connected in Reactome against a database of functional interactions. STRING [40–42] is a quality-controlled network of protein-protein associations from a diverse set of information, which aims to capture both physical interactions as well as functional relatedness. An interaction in STRING is assigned a confidence score based on seven evidence channels that range from experiment-based ("experiments" and "coexpression") to literature-based ("database" and "textmining") to genome-based channels ("neighborhood","fusion", and "co-occurrence"). Channel scores are converted to a single "combined score" for each interaction in STRING. This score and the individual channels provide a good benchmark dataset to evaluate whether scores are enriched for associations confirmed by Reactome (which may depend on the connectivity measure).

We ignored STRING channels that used transferred evidence via homology and the "database" channel, since Reactome was used as a source of evidence for this channel [42]. While the "combined score" channel includes information from Reactome, only 6% of the edges from this set have evidence from the "database" channel. For the remaining six channels and the "combined score" channel, we restricted our attention to the interactions where both proteins appear in the Reactome hypergraph representation. We then counted the number of interactions where both proteins appear in the same Reactome pathway (we considered a larger annotated set of 140 non-redundant Reactome pathways, see "Data formats and representations" in the Methods section). These interactions would typically be used when using a set-based approach for determining positive interactions. We then counted the number of interactions ($u$, $v$) where $u$ was connected to $v$ in the bipartite graph representation. In the combined score channel, 23% of the 1.4 million interactions appear in the same pathway and 14% interactions are connected in the bipartite graph (Fig 7A). However, 52% of the interactions in the same pathway were not connected and 19% of the connected interactions were not annotated to the same pathway, revealing that these two Reactome-based measures are not necessarily consistent. Venn diagrams for all channels exhibited a similar pattern (Fig 7, S8 and S9 Figs). Thus, a substantial number of STRING interactions contain nodes that are not annotated to the same pathway but are connected in the bipartite graph.

We sought to examine how the enrichment of interaction scores from STRING channels varied as we changed the $B$-relaxation distance between proteins, starting with the two extremes of bipartite graph connectivity and hypergraph $B$-connectivity. The distribution of interaction scores in different Reactome sets are dramatically different in the combined score channel (Fig 7A). The median score increases from 245 for all 1.4 million interactions to 541 for interactions within the same Reactome pathway. While the median score of connected nodes is relatively similar (at 624), these distributions are statistically different ($p < 0.01$ by the Kruskal-Wallis test). Strikingly, interactions where the nodes are $B$-connected within the hypergraph have a very large median score of 941; these are likely direct interactions within Reactome since $B$-connectivity is such a strict measure. As before, $B$-relaxation distance provides a smooth transition from $B$-connectivity to connectivity in the bipartite graph; selected thresholds are shown in the bottom panel of Fig 7A. B-connectivity preferentially connects the functionally-interacting pairs of proteins with the highest composite scores in STRING. We maintain this property up to a $B$-relaxation distance of 25, with a nearly three-fold increase in the number of interactions (67,751 to 180,373).

The "experimental" channel, which is comprised of IMEx interaction data [44], shows a similar pattern of statistically significant score enrichment with the more restrictive connectivity measures (Fig 7B). In this channel, $B$-relaxation distance preserves the enrichment of high-scoring interactions until a distance of about seven (S8 Fig). The "textmining" channel, which weights interactions based on protein co-mention in abstracts and other text collections [41], was also significantly enriched (Fig 7C). While the scores for the "textmining" channel remain

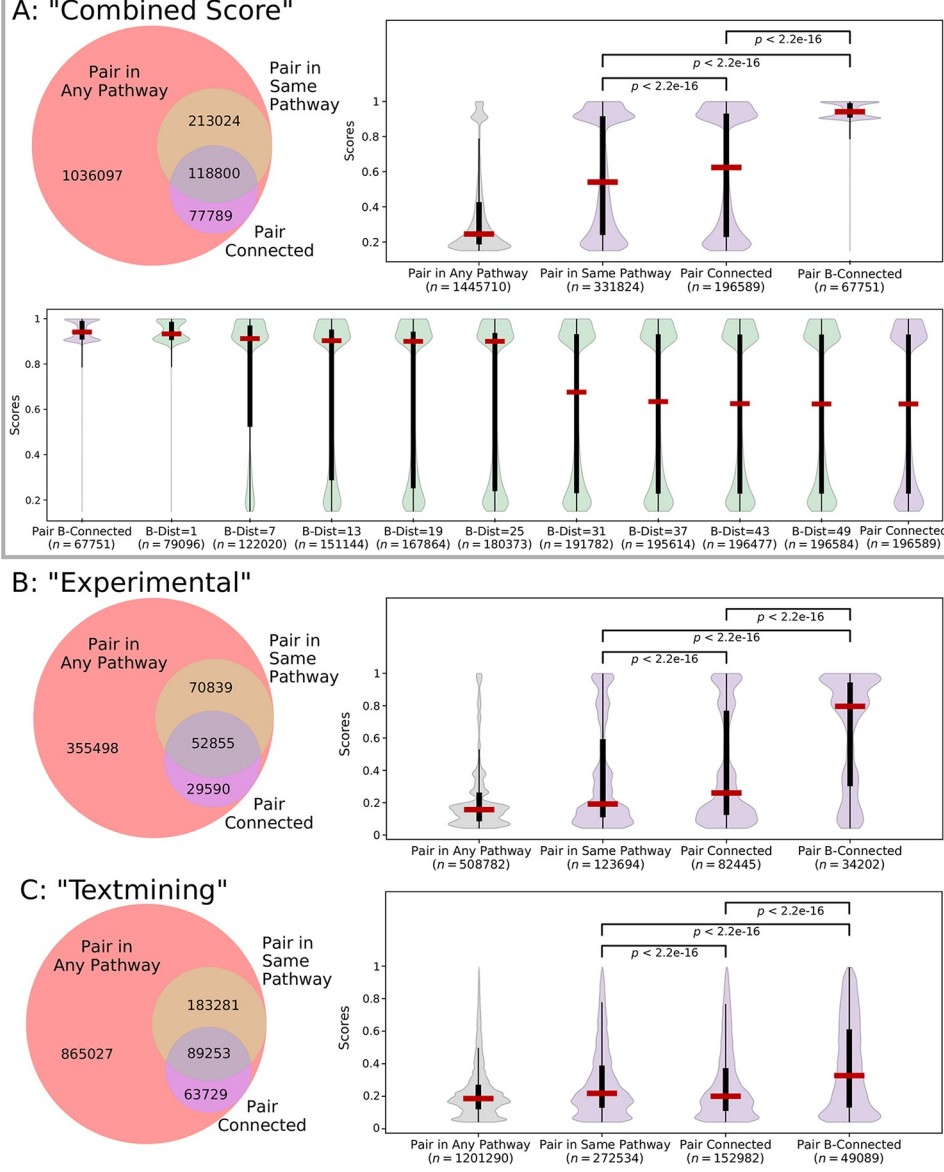

**Fig 7. STRING interactions within Reactome for (A) "combined score" interactions, (B) "experimental" interactions, and (C) "textmining" interactions.** The Venn diagram shows the overlap of interactions where the nodes appear in any Reactome pathway, appear in the same Reactome pathway, or are connected in the bipartite graph. The violin plot shows the distributions of interaction scores (which range from 1 to 1000) for different sets of interactions (median, percentiles, and Kruskal-Wallis *p*-values shown, where *p* < 2.2e − 16 is below the detection limit [43]). In Panel (A), the bottom violin plot shows the distributions of interaction scores for selected *B*-relaxation distance thresholds.

relatively low (the median hovers around 200 for pairs in any pathway), there are textmining scores for over 1.2 million interactions, and nearly 50,000 of these interactions are *B*-connected. The distribution of scores for other channels are shown in S9 Fig. Notably, the "coexpression" channel, which records correlation across gene expression datasets, is also improved slightly but significantly when considering stricter measures of connectivity. The genome-based "co-occurrence" and "fusion" channels did not have significant score enrichment across

the sets ($p \geq 0.01$). This was not surprising, since the genome-based channels are typically appropriate for Bacteria and Archea [41].

## Discussion

Connectivity is a foundational concept in cellular reaction networks, since it lies at the heart of determining the effect of one molecule upon another. The formal definition of connectivity is familiar and straightforward in directed graphs, the most common mathematical representation of reaction networks. However, precisely capturing this concept is challenging in more sophisticated and biologically accurate representations such as compound graphs, bipartite graphs, and directed hypergraphs. In recent years, scientists have developed these definitions independently for each of these representations.

This work is the first to systematically compare the relevant formulations of connectivity in four different models of reactions in signaling pathways. We study their impact on the Reactome database. We find that the directed graph representation of Reactome is very highly connected (90% of the nodes reach over 80% of the graph), the compound and bipartite graph versions are somewhat less connected (fewer than half the nodes reach less than half of the network), whereas the directed hypergraph model exhibits very poor connectivity (only five nodes are connected to more than 20 nodes). We attribute this trend to multiple, related factors. The SIF format for Reactome, from which we construct the directed graph, does not distinguish between modified forms of a protein and represents complexes as cliques. Compound graphs, bipartite graphs, and directed hypergraphs create a node for each form of a protein and for each protein complex. However, compound and bipartite graphs are much more connected than hypergraphs since they record multi-way reactions using multiple, independent edges. Directed hypergraphs accurately represent reactions, but their biologically-meaningful definition of connectivity (*B*-connectivity) is very restrictive in practice.

Motivated by these findings, we have provided a relaxed version of hypergraph connectivity, *B*-relaxation distance, that is tailored for the analysis of signaling pathways. *B*-relaxation distance takes the intuitive mechanical significance of *B*-connectivity and grants it the leeway necessary to deal with the challenges presented by the topologies of biomolecular hypergraphs. We show that *B*-relaxation distance elegantly bridges the gap between bipartite graphs and hypergraphs. Our algorithm for *B*-relaxation distance runs in polynomial time, and is efficient in practice. However, using directed hypergraphs to solve other computational problems can come with additional algorithmic challenges. For example, the shortest path problem on graphs is widely known to be solvable in polynomial time, while the analogous problem on directed hypergraphs is NP-complete [11, 28], even when bounding the number of nodes in the tail and head sets [29].

We use *B*-relaxation distance to identify downstream influence between annotated pathways in Reactome, defining an influence score $s_k$ that suggests how much a target pathway $T$ might be influenced by the downstream effects of a source pathway $S$. After performing an all-vs-all comparison across 34 Reactome pathways, we demonstrate the ability of *B*-relaxation distance to capture points of influence in two case studies: (a) the effect of the Mst1 pathway on MET signaling and (b) the role of Activin and BMP pathways on TGF$\beta$ signaling. These relationships are not recovered in the directed graph representation of Reactome. While the relationships are recovered for bipartite graphs, the value of $k$ is much larger than the best influence score based on *B*-relaxation distance. This reinforces the idea that *B*-relaxation distance is designed to quantify the amount of relaxation in the *B*-connectivity constraints, not necessarily the number of reactions that connect two molecules. Visualizing the hypergraph that contains nodes with small *B*-relaxation distance can pinpoint the exact reaction or

reactions responsible for the influence of one pathway on another. While our findings are not biologically novel, they demonstrate how researchers may explore Reactome in a systematic, unbiased manner to identify possible points of influence among pathways.

Our definitions of connectivity in Reactome may also identify functional relatedness of proteins, in addition to physical interactions. We used the STRING database as a benchmark dataset to evaluate the enrichment of scored interactions that are connected in Reactome. We found that many interactions are connected in Reactome but are not annotated to the same Reactome pathway, thus these potential downstream effects may be missed from gene-set-like enrichment approaches. Further, connectivity depends on the topology of Reactome and not the pathway annotations, so our approach may capture relationships that have not yet been co-annotated in Reactome. *B*-connectivity contains high-scoring interactions compared to bipartite connectivity for many evidence channels, and is most pronounced for the combined score that includes all channels. Thresholding by *B*-relaxation distance enables the addition of more interactions while still maintaining the score enrichment.

Computable representations of signaling pathways have been growing, and now Reactome and other databases contain details about reactions that are mined from the literature. However, even the state-of-the-art representations may fail to capture the full complexity of what we know about signaling, and there are certainly missing relationships among pathway molecules that have yet to be explored. *B*-relaxation distance and similar topology-based definitions are timely given the current state of pathway representations—they are more rigorous than graph-based or gene set-based measures, however they are not yet equipped to capture the intricacies of cellular signaling. For example, some descriptions of directed hypergraphs for signaling pathways represent complexes as hypernodes (node sets) [11], but this additional information is hard to reason about in terms of connectivity without manually-determined rules such as the CommonStream algorithm [25]. Appropriately modeling protein complexes will be essential to develop a method that is closer to capturing the "true" connectivity of a pathway. *B*-relaxation distance is an important first step in this direction.

As pathway databases such as Reactome continue to expand, *B*-relaxation distance will become a useful measure for systematically characterizing connectivity and relationships among annotated pathways. These reaction-centric databases also invite the generalization of other classic graph algorithms that have been used in biological applications to directed hypergraphs; in fact, random walks [45] and spectral clustering [46] have already been developed for directed hypergraphs with applications to other fields. Further, *B*-relaxation distance will allow us to re-examine potential correlations from experimental data (e.g. protein or gene expression), especially from perturbation experiments, using the topology provided by Reactome and other pathway databases.

## Methods

### Connectivity measures

Given a pathway and two entities, we wish to ask a very fundamental connectivity question: "is *a* downstream of *b*"? The answer to this question in directed graphs can be efficiently computed using a traversal algorithm such as breadth first search. Established connectivity measures on compound graphs [25] and hypergraphs [28] generalize breadth-first traversal. We begin with hypergraph connectivity and then describe our proposed relaxation to this measure, which is the main computational contribution in this work. We then describe another version of connectivity for compound graphs, which lies conceptually between graph connectivity and hypergraph connectivity.

**Hypergraph connectivity.** A directed hypergraph $\mathcal{H} = (V, \mathcal{E})$ contains a set $V$ of nodes and a set $\mathcal{E}$ of *hyperedges*, where a hyperedge $e = (T_e, H_e) \in \mathcal{E}$ consists of a tail set $T_e \subseteq V$ and a head set $H_e \subseteq V$ of nodes [28]. The *cardinality* of hyperedge $e$ is the sum of the nodes in the tail and head, i.e., $|T_e| + |H_e|$. Note that directed graphs are a special case of directed hypergraphs where $|T_e| = |H_e| = 1$ for each hyperedge $e$. In a directed graph, the set of nodes connected to some source $s$ is simply all nodes that are reachable via a path from $s$. The equivalent notion in a directed hypergraph is *B-connectivity*. Given a set of nodes $S \subseteq V$, *B-connectivity* ensures the property that traversing a hyperedge $\in \mathcal{E}$ requires that all the nodes in $T_e$ are connected to $S$. The following definition is adapted from Gallo et al. [28]:

**Definition 1**. Given a directed hypergraph $\mathcal{H} = (V, \mathcal{E})$ and a source set $S \subseteq V$, a node $u \in V$ is **B-connected** to $S$ if either (a) $u \in S$ or (b) there exists a hyperedge $e = (T_e, H_e)$ where $u \in H_e$ and each element in $T_e$ is B-connected to $S$. We use $B(\mathcal{H}, S)$ to denote the set of nodes that are B-connected to $S$ in $\mathcal{H}$.

We can compute $B(\mathcal{H}, S)$ using a hypergraph traversal [28]. This traversal works by finding hyperedges that have tails whose nodes are all B-connected to $S$, augmenting the set of B-connected nodes with the nodes in the heads of these hyperedges, and repeating this process until it does not discover any new nodes. The running time of this algorithm is linear in the size of $\mathcal{H}$.

**Parameterized hypergraph connectivity.** While *B-connectivity* is a biologically useful notion of connectivity, it is overly restrictive for the purpose of assessing the connectivity of pathway databases. We establish a relaxation of *B-connectivity* which works around such restrictions. Before we formally define *B-relaxation distance*, we distinguish different sets of hyperedges based on their association with the source set $S$ (Fig 8).

1. Given a hypergraph $\mathcal{H} = (V, \mathcal{E})$ and a source set $S \subseteq V$, a hyperedge $e = (T_e, H_e)$ is **reachable** from $S$ if at least one element of $T_e$ is B-connected to $S$.

2. Given a hypergraph $\mathcal{H} = (V, \mathcal{E})$ and a source set $S \subseteq V$, a hyperedge $e = (T_e, H_e)$ is **traversable** from $S$ if all elements of $T_e$ are B-connected to $S$.

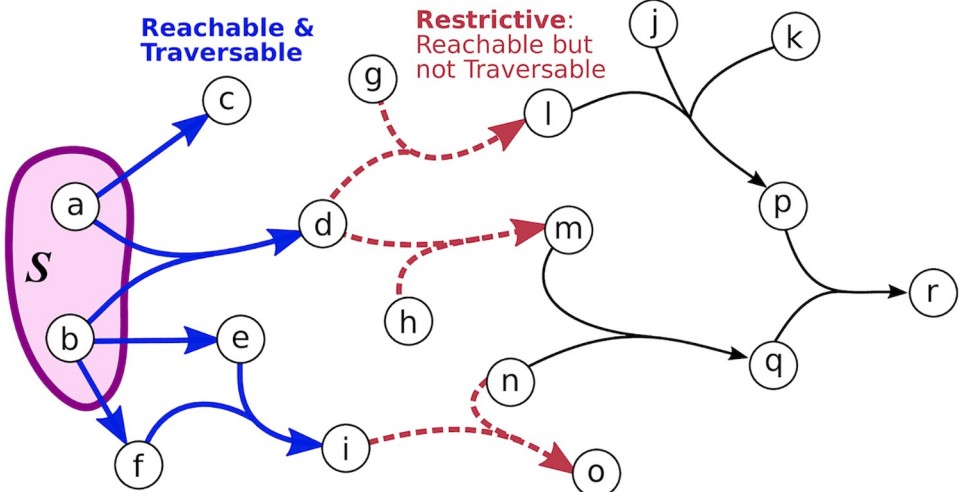

**Fig 8. Reachable, traversable and restrictive hyperedges.** This hypergraph has eight reachable hyperedges with respect to $S$: five traversable hyperedges (blue) and three restrictive hyperedges (red).

3. Given a hypergraph $\mathcal{H} = (V, \mathcal{E})$ and a source set $S \subseteq V$, a hyperedge $e$ is **restrictive** (with respect to $S$) if it is reachable but not traversable from $S$. We use $R(\mathcal{H}, S)$ to denote the set of restrictive hyperedges.

We modify the `b_visit()` algorithm from [28] to return the $B$-connected set $B(\mathcal{H}, S)$ and the restrictive hyperedges $R(\mathcal{H}, S)$ (Algorithm 1). The main difference between this traversal and a typical BFS is that a hyperedge is traversed only when all the nodes in the head have been visited. We also return the set of traversed hyperedges to avoid redundant computation in the relaxation algorithm that we describe later.

**Algorithm 1** `b_visit(`$\mathcal{H} = (V, \mathcal{E}), S \subset V$`)`

```
1: c[e] ← 0 for each hyperedge e ∈ ℰ        // counter of reached nodes
in e's tail
2: B ← S        // set of B-connected nodes
3: X ← ∅        // set of traversed hyperedges
4: Q ← S        // queue of nodes to traverse
5: while Q is nonempty do
6:   select and remove some node v ∈ Q
7:   for each hyperedge e ∈ ℰ where v ∈ Tₑ do
8:     c[e] ← c[e] + 1
9:     if c[e] = |Tₑ| then
10:        Q ← Q ∪ [Hₑ \ B]     // add unvisited heads of e to queue
11:        B ← B ∪ Hₑ         // add heads of e to B-connected set
12:        X ← X ∪ {e}        // add e to traversed hyperedges
13: R ← ∅        // set of restrictive hyperedges
14: for each hyperedge e ∈ ℰ do
15:   if c[e] ≥ 1 and c[e] < |Tₑ| then
16:      R ← R ∪ {e}        // hyperedge e reached but not traversed
   return B, R, X
```

We iteratively relax the notion of $B$-connectivity by allowing restrictive hyperedges to be traversed; to do so, at each iteration $k$ we need to keep track of $B_k(\mathcal{H}, S)$, the connected nodes, and $R_k(\mathcal{H}, S)$, the restrictive hyperedges. We initialize these sets to be the outputs of `b_visit()`:

$$B_0(\mathcal{H}, S) = B(\mathcal{H}, S) \tag{2}$$

$$R_0(\mathcal{H}, S) = R(\mathcal{H}, S). \tag{3}$$

In the $k$th iteration of this relaxation process, we consider the heads of each restrictive hyperedge $e$ from the previous iteration. $B_k(\mathcal{H}, S)$ is the set of $B$-connected nodes and $R_k(\mathcal{H}, S)$ is the set of restrictive hyperedges for each head set from $R_{k-1}(\mathcal{H}, S)$:

$$B_k(\mathcal{H}, S) = \bigcup_{e \in R_{k-1}(\mathcal{H}, S)} B(\mathcal{H}, H_e) \tag{4}$$

$$R_k(\mathcal{H}, S) = \bigcup_{e \in R_{k-1}(\mathcal{H}, S)} R(\mathcal{H}, H_e). \tag{5}$$

Note that computing $R_k(\mathcal{H}, S)$ using this definition requires $|R_{k-1}(\mathcal{H}, S)|$ different `b_visit()` calls, which is necessary to ensure that only one restrictive hyperedge is used to establish connectivity. With these definitions in hand, we are now ready to define our relaxation of $B$-connectivity.

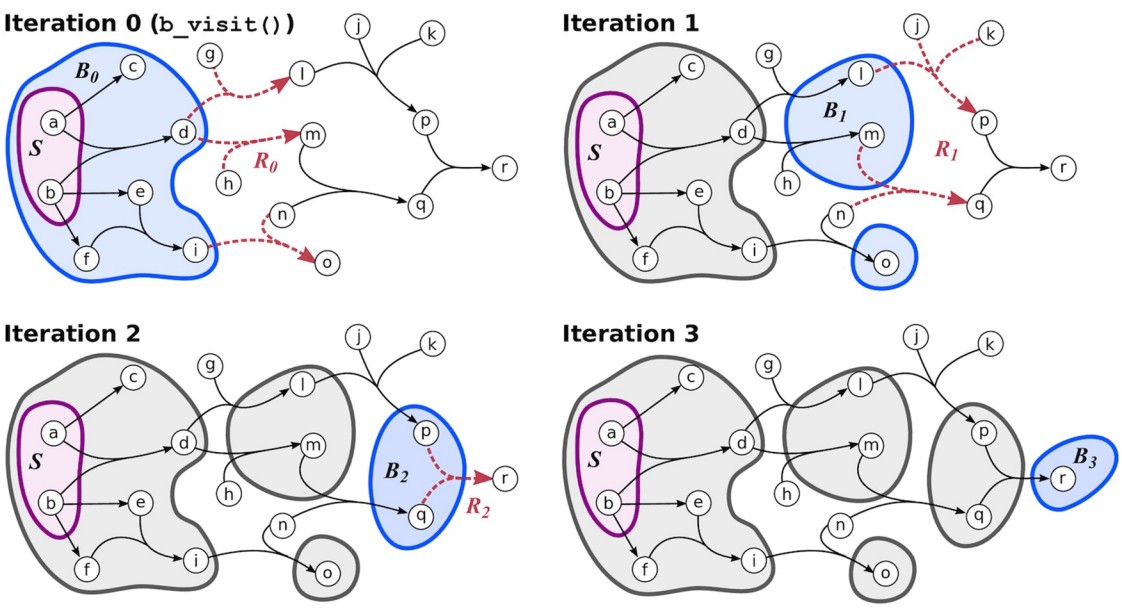

**Fig 9. Computing B-relaxation distance.** Connected nodes are in blue and restrictive hyperedges are in red for each iteration $k$. In this example, all nodes in gray are $B_3$-connected to $S = \{a, b\}$ and node $r$ has $B$-relaxation distance of three.

**Definition 2**. Given a hypergraph $\mathcal{H} = (V, \mathcal{E})$, a source set $S \subseteq V$, and an integer $k \geq 0$, a node $v \in V$ is $\boldsymbol{B_k}$**-connected** to $S$ if $v \in B_i(\mathcal{H}, S)$ for $i = 0, 1, \ldots, k$.

The $B$-*relaxation distance* of a node $v$ from a source set $S$ is the smallest value of $k$ such that $v$ is $B_k$-connected to $S$ in $\mathcal{H}$. In the main text, we use $B_{\leq k}$ to denote the $B_k$-connected set. An example of computing $B$-relaxation distance for all nodes in a hypergraph is shown in Fig 9. There may be different sets of hyperedges by which a node $v$ is $B_k$-connected to $S$ (S11 Fig).

**Algorithm 2** b_relaxation ($\mathcal{H} = (V, \mathcal{E}), S \subseteq V$)

```
 1: B₀, R₀, X ← b_visit(H, S)
 2: dist[v] ← 0 if v ∈ B₀ else ∞ for each node v ∈ V
 3: seen[e] ← True if e ∈ X else False for each hyperedge e ∈ E
 4: k ← 1
 5: while there exists some e ∈ R_{k-1} where seen[e] = False do
 6:    B_k ← ∅, R_k ← ∅
 7:    for e ∈ R_{k-1} where seen[e] = False do
 8:      seen[e] ← True
 9:      B, R, X ← b_visit(H, H_e)
10:      B_k ← B_k ∪ B
11:      R_k ← R_k ∪ R
12:      for e′ in X do
13:        seen[e′] ← True
14:    for v in B_k do
15:      if dist[v] = ∞ then
16:        dist[v] ← k
17:    k ← k + 1
18: return dist
```

We calculate the $B$-relaxation distance from $S$ to every node in the hypergraph by calling b_visit() on restrictive hyperedges for $k = 0, 1, 2, \ldots$ (Algorithm 2). Here, we declutter notation by dropping the parameterization of $\mathcal{H}$ and $S$ from the $B$-connected node set and the restrictive hyperedge set. The algorithm first calls b_visit() from $S$ to get the $B$-connected set $B_0$, the restrictive hyperedges $R_0$, and the traversed hyperedges $X$ (line 1). The $B$-relaxation

distance dictionary `dist` is initialized to 0 for nodes in $B_0$ and infinity otherwise, and the `seen` dictionary of hyperedges set to `True` if they have been traversed and `False` otherwise. While there are unseen restrictive hyperedges to traverse, the algorithm computes $B_k$ and $R_k$ by calling `b_visit()` on the heads of each restrictive hyperedge from iteration $k − 1$ (lines 7–11). We update the `seen` dictionary with all traversed hyperedges from each `b_visit()`, since these hyperedges may be restrictive with respect to another set of nodes and would be recomputed at a later iteration (lines 12–13, S10 Fig). Finally, the algorithm updates the `dist` dictionary for all nodes that are reached in the $k$-th iteration and increments $k$ (lines 14–17). This implementation keeps track of $B_0, B_1, \ldots, B_k$ and $R_0, R_1, \ldots, R_k$, which may be returned for other purposes.

**Runtime analysis.** The original `b_visit()` from Gallo et al. runs in $O(size(\mathcal{H}))$ time where $size(\mathcal{H})$ refers to the sum of the hyperedge cardinalities in $\mathcal{H}$ [28]. The modified `b_visit()` incurs no additional asymptotic runtime cost since the timing of the additional operations it conducts (Algorithm 1, lines 13–16) is trivially bounded by $|\mathcal{E}|$, which is bounded by $size(\mathcal{H})$.

In Algorithm 2, initializing the `dist` and `seen` dictionaries takes $|V|$ and $|\mathcal{E}|$ time, respectively. The while loop (line 5) contains two for loops. The first loop in line 7 iterates over all restrictive hyperedges, performing work only when that hyperedge has not been previously traversed. Thus, the code in the first loop will be executed at most $|\mathcal{E}|$ times over the full course of the algorithm, corresponding to the case where every hyperedge in $\mathcal{H}$ appears in some restrictive set. The first loop calls `b_visit()` in line 9 at each iteration, which runs in $O(size(\mathcal{H}))$ time as previously mentioned. The second loop in line 14 updates the $B$-relaxation distance of each node exactly once, when it is first discovered by the algorithm. It will be executed at most $|V|$ times over the full course of the algorithm. The running time of the first loop (line 7) dominates those of the initialization steps and the distance update loop; thus, the runtime of Algorithm 2 is $O(|\mathcal{E}| \cdot size(\mathcal{H}))$.

**Pre-processing speedup.** When we ran `b_relaxation()` on each source node on the Reactome hypergraph, the algorithm took an average of 31.6 seconds per node on a Linux machine with quad Intel Core i7-4790 processors. The quadratic runtime is tractable for a handful of calls, but calling `b_relaxation()` from every vertex in $V$ (as we do in this work) will result in a cubic runtime. We formulated an optimized version of `b_relaxation()`, which we initialized by calling `b_visit()` on $H_e$ for each $e \in \mathcal{E}$ and recording the resulting connected nodes and restrictive hyperedges. This initialization step incurs a cost of $|\mathcal{E}| \cdot size(\mathcal{H})$ time, but replaces the call to `b_visit()` in line 9 with a constant-time lookup operation. Thus the sole quadratic term in the runtime of Algorithm 2 becomes linear in the optimized version. The optimized version, when applied to each source node on the Reactome hypergraph, gave an average running time of 0.310 seconds per node, giving an improvement of two orders of magnitude.

**Compound graph connectivity.** There are multiple definitions of compound graphs [8, 25]. Here we describe *compound pathway graphs CP = (G, I)* that consist of two graphs [25]. The pathway graph $G = (V, E_G)$ is a mixed graph where $V$ denotes the set of nodes and $E_G$ denotes the interaction and regulation edges among nodes, some of which may be directed. Edges may also denote inhibition/activation; here, we ignore this aspect of the compound graph. The inclusion graph $I = (V, E_I)$ is on the same node set $V$ and $E_I$ denotes the undirected edges for defining compound structure membership (e.g., complexes and abstractions). To traverse a compound pathway graph, we need, for each compound structure, two flags: (a) `compound`: if a compound structure is reached, are all its members also reached? and (b) `member`: if a member of a compound structure is reached, are all other members in the compound structure also reached? During the traversal, once a node $u$ is reached, the

algorithm determines if any other nodes are "equivalent" to *u* based on these flags. Note that while compound graphs handle traversals through entities such as protein complexes and families, the edges only connect pairs of these entities. Thus, the requirements imposed by *B*-connectivity on hypergraphs cannot be implemented on compound graphs as they are currently defined.

A *compound path* between two nodes consists of edges that are either from the pathway graph $E_G$ or represent a link between nodes that are equivalent for traversal based on the `compound` and `member` flags. These compound paths are used to establish the set of nodes that are downstream of a source node. For comparison with other measures, we modify the definition from [25] to ignore activation/inhibition effects and remove a restriction on path lengths:

**Definition 3**. Given a compound pathway graph $CP = (G, I)$ and a source set $S \subseteq V$, a node $u \in V$ is **downstream** of *S* in *CP* if there exists some compound path from any node $s \in S$ to *u* in *CP*.

We run the `DOWNSTREAM` algorithm implemented in the PaxTools software [25, 47] on each source node in *S*, ignoring activation/inhibition sign and the path length limit.

## Data formats and representations

We automatically generate the four Reactome representations—directed graph, compound graph, bipartite graph, and hypergraph—using a suite of tools (Fig 10). We use PathwayCommons, a unified collection of publicly-available pathway data [19], to collect BioPAX and SIF files representing the entire Reactome database (http://www.pathwaycommons.org/archives/PC2/v10/). The SIF files are generated by PathwayCommons by converting BioPAX relationships to binary relations; more details are available at http://www.pathwaycommons.org/pc2/formats. We convert the SIF files to a directed graph by converting each binary relation to a directed or bidirected graph (S2 Table).

We use the PaxTools Java parser to work with BioPAX files [47]. PaxTools offers querying algorithms such as `DOWNSTREAM` that operates on the compound graph representation [25]. We use PaxTools to construct hypergraphs by traversing the BioPAX files. For each biochemical

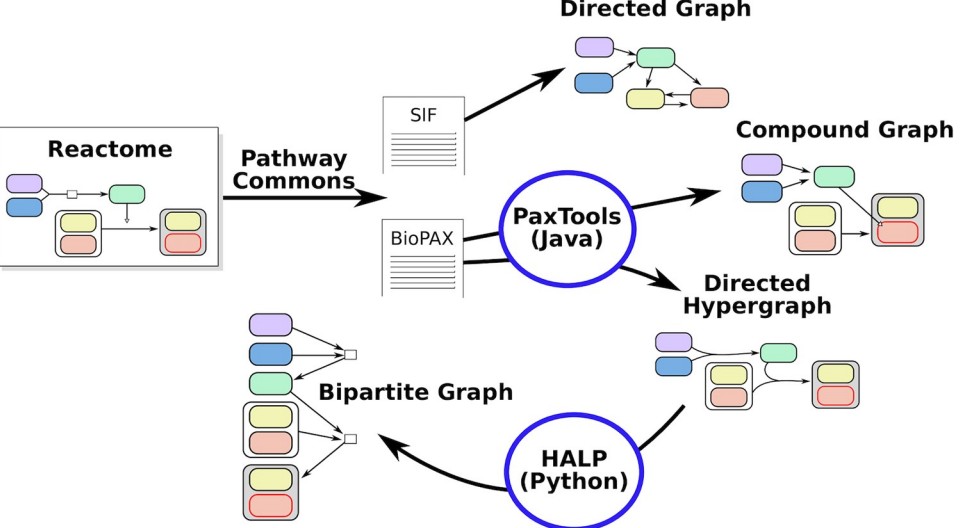

**Fig 10. Building pathway representations from Reactome.**

reaction in BioPAX, we construct a hyperedge with the reactants and control elements in the tail and the products in the head. We use the algorithms provided in the Hypergraph Algorithms Package (HALP, http://murali-group.github.io/halp/, GPL-3.0 license) to work with hypergraphs. The *B*-relaxation distance algorithm is provided in HALP's `annabranch`.

Finally, we build the bipartite graph directly from the hypergraph, converting each hyperedge *e* into a reaction node *r* and connecting the tails of *e* to *r* and then *r* to the heads of *e*. Thus, the number of nodes in the bipartite graph is exactly the number of nodes plus the number of hyperedges in the hypergraph, and large *B*-relaxation distance corresponds to traversing the bipartite graph.

Source code for preprocessing, connectivity survey, and experiments on hypergraph and other representations of Reactome are available on GitHub (https://github.com/annaritz/pathway-connectivity, GPL-3.0 license).

**Reactome pathways used for influence analysis and STRING benchmarking.** We consider 34 Reactome pathways for assessing pathway influence across reactome. Pathways are organized hierarchically within Reactome, and the 34 pathways were labeled as direct children of "Signal Transduction" and "Signaling by Receptor Tyrosine Kinases" in the Reactome file provided by PathwayCommons v10 (S1 Table).

When we benchmark functional relationships using STRING evidence channels, we expand the set of Reactome pathways to include others not annotated to "Signal Transduction." We identified 140 non-redundant pathways for this analysis as follows. We first collected entities annotated to each Reactome pathway and sub-pathway, and then removed the 26 "top-level" pathways in the Reactome hierarchy from consideration (Cell Cycle, DNA Repair, Immune System, etc.). These "top-level" pathways are general and are represented by more specific sub-pathways in the list. We finally removed all sub-pathways that are completely contained in another pathway. These redundant pathways do not add any additional pairwise relationships to the benchmarking analysis.

**Hypergraph visualization.** We visualize hypergraphs using GraphSpace [48], a web-based collaborative network visualization tool. The hypergraphs are available as interactive networks on GraphSpace using the with the `GLBio2019` tag (http://graphspace.org/graphs/?query=tags:glbio2019).

## Supporting information

**S1 Fig. Heatmaps of filtered pathway representations.** Heatmaps showing the effect of filtering pathway representations by blacklisted nodes and small molecules. (A) The proportion of nodes $|B_{\leq k}|$ in the $B_k$-connected set from each source node (rows) for values of $k$ (columns) in the hypergraph. (B) Directed graph connectivity, bipartite graph connectivity, and hypergraph $B$-connectivity for representations with blacklisted nodes removed. (C) Directed graph connectivity, bipartite graph connectivity, and hypergraph $B$-connectivity for representations with small molecules and three highly-connected entities (cytosolic Ubiquitin, nuclear Ubiquitin, and the Nuclear Pore Complex) removed.
(PDF)

**S2 Fig. Permutation test for pathway membership.** The example shows ten molecules $(x_1, \ldots, x_{10})$ that are members of three pathways (*A*, *B* and *C*). The initial pathway overlap, represented as a Venn diagram, results in the matrix of pairwise overlaps (left). We construct an undirected bipartite graph (we'll call this the *permutation graph* to distinguish this graph from the bipartite graph representation). In the permutation graph, one set of nodes are the molecules and the other set of nodes are all possible overlapping sets except the null set (here, $2^n - 1 = 2^3 - 1 = 7$). Edges in the permutation graph connect molecules to the overlapping set

to which they belong. We then perform degree-preserving edge swaps by selecting pairs of edges with different nodes and swapping them (we perform 10,000 swaps in all experiments). We then "piece" the Venn diagram back together, which contains the same number of elements in each overlapping set (resulting in the same pairwise overlaps). With 34 pathways, one would think that the permutation graph is too large; however, we found that there were only 255 non-empty portions of the Venn diagram for the hypergraph/bipartite graph entities (and only 180 for the directed graph entities). The size of the permutation graph also makes the choice of 10,000 swaps for each permutation reasonable.
(PDF)

**S3 Fig. Hypergraph influence scores.** Influence scores of pairs of Reactome pathways for the hypergraph at selected values of $B$-relaxation distance $k$. Rows indicate the source pathway $P_S$ and columns indicated the target pathway $P_T$. Color indicates influence score and circle size indicates significance by permutation test (larger circles are more significant). Three selected distances are enlarged (note $s_3$ is also in the main manuscript).
(PDF)

**S4 Fig. Directed graph influence scores.** Influence scores of pairs of Reactome pathways for the directed graph at selected values of distance $k$. Rows indicate the source pathway $P_S$ and columns indicated the target pathway $P_T$. Color indicates influence score and circle size indicates significance by permutation test (larger circles are more significant). Three selected distances are enlarged. Note that the entities are different for this graph than the hypergraph and bipartite graph, resulting in different initial pathway overlaps.
(PDF)

**S5 Fig. Bipartite graph influence scores.** Influence scores of pairs of Reactome pathways for the bipartite graph at selected values of distance $k$. Rows indicate the source pathway $P_S$ and columns indicated the target pathway $P_T$. Color indicates influence score and circle size indicates significance by permutation test (larger circles are more significant). Three selected distances are enlarged.
(PDF)

**S6 Fig. Bipartite graph pathway influence for Mst1, BMP, and Activin.** The influence of (A) signaling by Mst1, (B) signaling by BMP, and (C) signaling by Activin on the other Reactome pathways. The dashed black line indicates the number of nodes in the source pathway's $B_{\leq k}$ for different values of $k$. There is one line for each of the 33 other target pathways denoting the number of members that appear in $B_{\leq k}$, with selected pathways highlighted in bold.
(PDF)

**S7 Fig. Activin $B$-relaxation distance hypergraph.** Hyperedges traversed to compute $B_0$, $B_1, \ldots, B_4$ from source pathway Activin. Node colors represent $B$-relaxation distance from $k = 0$ ($B$-connected set, blue) to $k = 3$ (bright green). Gray nodes are entities that are not in the $B_k$-connected set but are involved in traversed hyperedges. Star-shaped nodes are members of the TGF$\beta$ pathway. This network is available on GraphSpace at http://graphspace.org/graphs/26756?user_layout=6713.
(PNG)

**S8 Fig. STRING "experimental" channel.** STRING interactions within Reactome for "experimental" interactions. In addition to the components of Fig 7, the bottom violin plot shows the distributions of interaction scores for selected $B$-relaxation distance thresholds.
(PDF)

**S9 Fig. Other STRING channels.** STRING interactions within Reactome for the remaining evidence channels not shown in Fig 7 or S8 Fig. The Venn diagram shows the overlap of interactions where the nodes appear in any Reactome pathway, appear in the same Reactome pathway, or are connected in the bipartite graph. The violin plot shows the distributions of interaction scores for different sets of interactions (median and percentiles shown; Kruskal-Wallis $p$-values less than 0.01 are shown with a solid line).
(PDF)

**S10 Fig. Restrictive hyperedges may include ones traversed in previous iterations.** The restrictive set $R_k$ may include hyperedges that have been traversed in a previous iteration's `b_visit()` call. In iteration 1, the restrictive set $R_1$ is established by considering the $B$-connectivity from the heads of the two hyperedges in $R_0$. The hyperedge $\{\{e, f\}, \{g\}\}$ is restrictive with respect to the heads of one hyperedge in $R_0$ but traversable with respect to the heads of the other hyperedge. Thus, $\{\{e, f\}, \{g\}\}$ is included in $R_1$ but also added to the `seen` dictionary, saving redundant computation in Algorithm 2.
(PDF)

**S11 Fig. $B$-relaxation distance example.** Examples of connectivity from $S = \{a, b\}$ to $r$ with a $B$-relaxation distance of three. Blue hyperedges denote traversals that are consistent with $B$-connectivity; red hyperedges denote traversals where one, but not all, nodes in the tail are connected; only hyperedges that are involved in the connectivity from $S$ to $r$ are highlighted for simplicity. Note that while $B$-relaxation distance is three, there are different sets of hyperedges that achieve this $B$-relaxation distance.
(PDF)

**S1 Table. Reactome signaling pathways considered for pathway influence analysis.** Members that are not part of any hyperedge are ignored from the hypergraph. The filtered hypergraph has removed all small molecules, two forms of Ubiquitinase, and the Nuclear Pore Complex from the hyperedges.
(PDF)

**S2 Table. Rules for converting SIF binary relations to directed edges.** We ignore the "neighbor-of" binary relation.
(PDF)

**S1 File. Influence score values.** Pathway overlaps, influence scores, and permutation test values for different values of $k$.
(XLSX)

## Acknowledgments

We thank Brendan Avent for his initial work on the hypergraph algorithms library and Ozgun Babur for discussions about BioPAX and PaxTools.

## Author Contributions

**Conceptualization:** Nicholas Franzese, Adam Groce, T. M. Murali, Anna Ritz.

**Data curation:** Nicholas Franzese, Anna Ritz.

**Formal analysis:** Nicholas Franzese, Adam Groce, T. M. Murali, Anna Ritz.

**Methodology:** T. M. Murali, Anna Ritz.

**Software:** Nicholas Franzese, Anna Ritz.

**Supervision:** Adam Groce, T. M. Murali, Anna Ritz.

**Validation:** Anna Ritz.

**Visualization:** Nicholas Franzese, Anna Ritz.

**Writing – original draft:** Nicholas Franzese, Adam Groce, T. M. Murali, Anna Ritz.

**Writing – review & editing:** T. M. Murali, Anna Ritz.

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
