## [Decision Letter · Decision Letter 0]

10 Jul 2019

Dear Dr Ritz,

Thank you very much for submitting your manuscript, 'Connectivity measures for signaling pathway topologies', to PLOS Computational Biology. As with all papers submitted to the journal, yours was fully evaluated by the PLOS Computational Biology editorial team, and in this case, by independent peer reviewers including two from the GLBIO submission. The reviewers agreed that this revision is responsive to the major concerns raised but identified some aspects of the manuscript that should be improved.

We would therefore like to ask you to modify the manuscript according to the review recommendations before we can consider your manuscript for acceptance. Your revisions should address the specific points made by each reviewer and we encourage you to respond to particular issues. Please note while forming your response, if your article is accepted, you may have the opportunity to make the peer review history publicly available. The record will include editor decision letters (with reviews) and your responses to reviewer comments. If eligible, we will contact you to opt in or out.raised.

- Supporting Information uploaded as separate files, titled 'Dataset', 'Figure', 'Table', 'Text', 'Protocol', 'Audio', or 'Video'.

We hope to receive your revised manuscript within the next 30 days. If you anticipate any delay in its return, we ask that you let us know the expected resubmission date by email at ploscompbiol@plos.org.

Sincerely,

Ferhat Ay, Ph.D

Associate Editor

PLOS Computational Biology

Jason Papin

Editor-in-Chief

PLOS Computational Biology

[LINK]

Reviewer's Responses to Questions

**Comments to the Authors:**

Reviewer #1: The authors have addressed my comments fully in the revised manuscript and response to reviewers. A very nice paper!

I have some minor typographical and cosmetic comments:

- The type size in the compiled manuscript in all figures except for 4 is too small. If possible, it would be nice to standardize the font size across the figures.

- I think that the cross-refs for Fig 3B and 3C might be flipped in the text.

- Figure 4A: Is it possible to stretch A so that one can look at the same pathway across both A and B?

- Figure 4B: What is the threshold for significance? Are the p-values corrected for multiple testing?

- In the Abstract: “pairs of pathways statistically significant influence” → “pairs of pathways with statistically significant influence”

- Page 3, line 43: k hasn’t actually been defined before this point

- Page 4, line 68: “built upon” – “encoded within”?

- Page 6, lines 145-162: I got a bit confused re-reading this section. The text describes three sets of nodes to remove: 2778 small molecules defined by Reactome, 3 highly connected gene products/complexed, and another 155 small molecules from PathwayCommons. Which of these are “blacklisted nodes” – the ubiquitins and NPC? Were both sets of small molecules combined into one? Do the stats on the numbers of affected nodes/hyperedges reflect the removal of all three sets of nodes?

- Page 10, line 242: “under any definition” -- definition of what?

- Page 11, line 307: “Reactome pathwa” → “Reactome pathway”

Reviewer #2: I was a reviewer for the conference version of the paper to GLBIO. The authors have done a great job in their response to reviewers. The manuscript is in great shape, and I don't have any major comments.

Minor comment: p.11 line 23: typo, pathwa  pathway

Reviewer #3: As a new reviewer for this revised submission, these comments are intended to assess only the responsiveness of the revisions to the major concerns from the original reviewers. The major changes in the revision include several new analyses that better demonstrate how B-relaxation distance can be used in practice and how this distance measure relates to existing connectivity measures. The new figures showing reachable source-target paths as a function of distance satisfy prior concerns about how the parameterized B-relaxation measure relates to parameterized versions of other connectivity measures.

Several reviewers questioned how B-relaxation can be evaluated empirically with more comprehensive applications. The new evaluation of STRING edges by score and edge type is a strong response to these comments. The results depict how pairs of proteins that are B-connected or have a particular B-relaxation distance tend to have higher STRING edge scores, especially for the STRING combined scores. This emphasizes how B-relaxation provides a different way to evaluate node connectivity that is not trivially recapitulated by the existing connectivity measures or Reactome pathway membership. The reviewers noted several additional potential applications that the authors defer for future work. This is reasonable to keep outside the scope of this manuscript and shows the opportunities for future studies based on B-relaxation connectivity. In addition, the authors added new permutation-based statistical testing for their Reactome pathway influence analysis. This is an appropriate control because the authors are correct that simulating realistic pathway hypergraphs, a previously-suggested alternative way to test the influence measures, is non-trivial.

The minor comments regarding the precision and recall analysis for the pathway influence study would have been a helpful addition to the manuscript. However, this is a fairly minor request and would not substantially affect the conclusions about the merits of B-relaxation.

Overall, all major concerns have been directly or indirectly addressed. The GraphSpace network visualizations and software to reproduce the major analyses in the manuscript are added benefits that complement the strong manuscript. The specific comments below are all minor comments that could help improve the presentation.

Minor comments:

1) Reactome has a large number of pathways, but only 34 are used for the pathway influence analysis and 140 are used for the STRING edge score analysis. It is unclear why these subsets where selected and why different pathways were used for the influence and STRING analyses.

2) The STRING analysis excluded the "database" channel, which used Reactome as a source of evidence. However, if the "combined score" channel includes the "database" channel as an information source, there is still some degree of circularity in this analysis.

3) The figures for the STRING analysis show scores in the 0-1000 range. However, some of the text describes scores in the 0-1 range.

4) Using 2.2e-308, the smallest float in Python, in the STRING significance analysis may not be the best way to present these results. Wolfgang Huber discussed this issue recently in "Reporting p Values" (Cell Systems https://doi.org/10.1016/j.cels.2019.03.001).

5) In Figure 8 and other figures that visualize hypergraph properties, it is not immediately obvious that the blue regions overlap the red regions. Even if the figures are not changed, it would be helpful to explicitly note in the caption that node d is in both the blue and the red sets.

6) The Discussion notes that connected proteins that are not in the same pathway may be missed from gene-set-like enrichment approaches. For most applications of gene set enrichment, this seems like a desirable property.

7) The Python software is provided in https://github.com/Murali-group/halp and https://github.com/annaritz/pathway-connectivity and is well-organized in general. However:

- The GPL3 licenses are not stated directly in the manuscript.

- It is not clear which developmental branch of the halp repository was used for this study, which impedes reproducibility. Ideally this version would be merged to master and released.

- There are test cases, but the Travis CI testing fails.

- The pathway-connectivity repository contains data from third-party sources (e.g. STRING) that may require more direct attribution in the readme.

- An added bonus for both repositories would be to archive the software on third-party services like Zenodo, Figshare, or Software Heritage to ensure they are more permanently available to readers.

8) Typos and phrasing:

- Abstract: "pathways statistically significant" is missing a word

- P3: "molecules that that participate"

- P3: The end of the introduction notes relationships that "emerge for larger values of k" but the parameter k has not yet been introduced and is not intuitive before that introduction

- P9: "nodes that in"

- P11: "within the same Reactome pathwa"

- P16: "PaxTools java" could capitalize Java

**Have all data underlying the figures and results presented in the manuscript been provided?**

Reviewer #1: No: The statistics underlying the figures are not provided as supplementary tables, but the authors have provided their input data and code to generate the results on GitHub.

Reviewer #2: Yes

Reviewer #3: Yes

PLOS authors have the option to publish the peer review history of their article (what does this mean?). If published, this will include your full peer review and any attached files.

Reviewer #1: No

Reviewer #2: No

Reviewer #3: No

---

## [Decision Letter · Decision Letter 1]

9 Sep 2019

Dear Dr Ritz,

We are pleased to inform you that your manuscript 'Hypergraph-based connectivity measures for signaling pathway topologies' has been provisionally accepted for publication in PLOS Computational Biology.

In the meantime, please log into Editorial Manager at https://www.editorialmanager.com/pcompbiol/, click the "Update My Information" link at the top of the page, and update your user information to ensure an efficient production and billing process.

One of the goals of PLOS is to make science accessible to educators and the public. PLOS staff issue occasional press releases and make early versions of PLOS Computational Biology articles available to science writers and journalists. PLOS staff also collaborate with Communication and Public Information Offices and would be happy to work with the relevant people at your institution or funding agency. If your institution or funding agency is interested in promoting your findings, please ask them to coordinate their releases with PLOS (contact ploscompbiol@plos.org).

Thank you again for supporting Open Access publishing. We look forward to publishing your paper in PLOS Computational Biology.

Sincerely,

Ferhat Ay, Ph.D

Associate Editor

PLOS Computational Biology

Jason Papin

Editor-in-Chief

PLOS Computational Biology

Reviewer's Responses to Questions

**Comments to the Authors:**

Reviewer #3: The authors have addressed all of my previous minor concerns with these revisions. I noted two trivial formatting issues in this version:

- "from" is repeated in line 46

- The "All Nodes" label cut off in Fig S6B

**Have all data underlying the figures and results presented in the manuscript been provided?**

Reviewer #3: Yes

PLOS authors have the option to publish the peer review history of their article (what does this mean?). If published, this will include your full peer review and any attached files.

Reviewer #3: No

---

## [Editor Report · Acceptance letter]

16 Oct 2019

PCOMPBIOL-D-19-00839R1 

Hypergraph-based connectivity measures for signaling pathway topologies

Dear Dr Ritz,

I am pleased to inform you that your manuscript has been formally accepted for publication in PLOS Computational Biology. Your manuscript is now with our production department and you will be notified of the publication date in due course.

With kind regards,

Matt Lyles
